# Carvedilol improves glucose tolerance and insulin sensitivity in treatment of adrenergic overdrive in high fat diet-induced obesity in mice

Linh V. Nguyen[1], Quang V. Ta[2], Thao B. Dang[1¤a], Phu H. Nguyen[3], Thach Nguyen[1¤b], Thi Van Huyen Pham[1], Trang HT. Nguyen[4], Stephen Baker[4], Trung Le Tran[5], Dong Joo Yang[5], Ki Woo Kim[5]*, Khanh V. Doan[1]*

1 School of Medicine, Tan Tao University, Long An, Viet Nam, 2 School of Biotechnology, Tan Tao University, Long An, Viet Nam, 3 Binh Dan Hospital, Ho Chi Minh, Viet Nam, 4 Oxford University Clinical Research Unit in Viet Nam, Ho Chi Minh, Viet Nam, 5 Division of Physiology, Department of Oral Biology, BK21 PLUS, Yonsei University College of Dentistry, Seoul, South Korea

¤a Current address: Department of Internal Medicine, Texas Tech University Health Science Center at the Permian Basin, Odessa, TX, United States of America.
¤b Current address: Department of Cardiology, Methodist Hospital, Merrillville, IN, United States of America.
* KIWOO-KIM@yuhs.ac (KWK); khanh.doan@ttu.edu.vn (KVD)

**Data Availability Statement:** All relevant data are within the paper and Supporting Information files.

## Abstract

Catecholamine excess reflecting an adrenergic overdrive of the sympathetic nervous system (SNS) has been proposed to link to hyperleptinemia in obesity and may contribute to the development of metabolic disorders. However, relationship between the catecholamine level and plasma leptin in obesity has not yet been investigated. Moreover, whether pharmacological blockade of the adrenergic overdrive in obesity by the third-generation beta-blocker agents such as carvedilol could help to prevent metabolic disorders is controversial and remains to be determined. Using the high fat diet (HFD)-induced obese mouse model, we found that basal plasma norepinephrine, the principal catecholamine as an index of SNS activity, was persistently elevated and highly correlated with plasma leptin concentration during obesity development. Targeting the adrenergic overdrive from this chronic norepinephrine excess in HFD-induced obesity with carvedilol, a third-generation beta-blocker with vasodilating action, blunted the HFD-induced hepatic glucose over-production by suppressing the induction of gluconeogenic enzymes, and enhanced the muscular insulin signaling pathway. Furthermore, carvedilol treatment in HFD-induced obese mice decreased the enlargement of white adipose tissue and improved the glucose tolerance and insulin sensitivity without affecting body weight and blood glucose levels. Our results suggested that catecholamine excess in obesity might directly link to the hyperleptinemic condition and the therapeutic targeting of chronic adrenergic overdrive in obesity with carvedilol might be helpful to attenuate obesity-related metabolic disorders.

**Funding:** This research was funded by the Viet Nam National Foundation for Science and Technology (NAFOSTED 108.05-2017.01 to KVD), URL: https://nafosted.gov.vn/en/. The funders had no role in study design, data collection and analysis, decision to publish, or preparation of the manuscript.

**Competing interests:** The authors have declared that no competing interests exist.

## Introduction

Obesity and related adverse consequences such as metabolic disorders and cardiovascular diseases (CVDs) have become a global health problem in modern society [1]. Overactivation of the sympathetic nervous system (SNS) which has been well-documented in obesity [2] plays an important role in the pathogenesis of obesity-associated CVDs [3, 4] and may also contribute to the development of metabolic disorders in obesity [5–7]. Compelling evidence has suggested that the hyperleptinemic condition (increased plasma leptin concentration) in obesity is one of important factors contributing to the hyper-activated SNS which eventually leads to many cardiovascular complications [3, 8–10].

Recent studies have shown that the metabolic and cardiovascular effects of leptin might be mediated via an increase in catecholamine signaling [11–14], possibly via modulating the synthesis and release of catecholamine [15–21]. As final mediators of the SNS activity, catecholamine plays a crucial role in the physiology of neurotransmission, physical and mental activities, cardiovascular function, metabolism, inflammation and immunity [22–24]. However, catecholamine excess which leads to adrenergic receptor overactivation is known to cause profoundly adverse effects on metabolism and cardiovascular function as seen in patients with pheochromocytoma [25, 26]. Nonetheless, whether catecholamine excess is observed in obesity and links to hyperleptinemia has not yet been investigated. Moreover, it remains controversial whether targeting the SNS hyperactivity in obesity by pharmacological treatment could help to prevent metabolic disorders [6].

Beta-blockers, available therapeutic agents in clinical treatment of many cardiovascular diseases, competitively antagonize endogenous catecholamine on β-adrenergic receptors of the SNS [27, 28]. Beta-blockers are classified into first, second, and third generation depending on their $\beta_1/\beta_2$-adrenoceptor selectivity and intrinsic vasodilatory properties [27]. Even though it seems reasonable that blocking the SNS hyperactivity in patients with obesity/metabolic syndrome might provide metabolic benefits, the clinical use of traditional beta-blockers (first and second generation agents) such as atenolol in these patients is concerning due to their negative metabolic effects including glucose intolerance, dyslipidemia, and inability to lose weight [29, 30]. It has been suggested that the third-generation beta-blockers which possess vasodilating properties either by an additive α-adrenoceptor antagonism or a nitric oxide-synthesizing stimulation provide complete sympathetic blockade and may mitigate the negative metabolic effects of traditional beta-blockers [31]. This notion was supported by results from recent clinical studies showing neutral to favorable effects of the third-generation beta-blockers on metabolic profiles compared to traditional beta-blockers [32, 33]. However, molecular metabolic effects on the glucose tolerance and insulin sensitivity of these newer beta-blockers in obesity have not been investigated.

In the present study, we found that basal plasma norepinephrine, the principal catecholamine as an index of SNS activity [34], was persistently elevated and highly correlated with the plasma leptin concentration, but not plasma insulin, during high fat diet (HFD)-induced obesity in mice. Targeting the adrenergic overdrive from this chronic norepinephrine excess in HFD-induced obesity with carvedilol, a third-generation beta-blocker agent with additive $\alpha_1$-adrenoceptor antagonizing action [35, 36], blunted hepatic glucose overproduction by suppressing the induction of gluconeogenic enzymes and increased the muscular insulin signaling pathway. These metabolic effects eventually led to an improvement in glucose tolerance and insulin sensitivity. These results suggested a high correlation between chronic norepinephrine excess and hyperleptinemia in obesity and the therapeutic treatment of chronic adrenergic overdrive with carvedilol might be helpful to attenuate the glucose and insulin intolerance associated with obesity.

## Materials and methods

### Antibodies and reagents

Primary antibodies including Akt (Cat. No. 2920, dilution 1:10,000), p-Akt (Cat. No. 3787, dilution 1:10,000), Creb (Cat. No. 9197, dilution 1:2,000), p-Creb (Cat. No. 9196, dilution 1:2,000), PEPCK1 (Cat. No. 12940, dilution 1:2,000) and p-IGF-1Rβ/ InsRβ (Cat. No. 3021, dilution 1:2,000) were obtained from Cell Signaling (Cell Signaling Technology Inc., MA, USA). G6Pase (Cat. No. ab83690, dilution 1:2,000), PPARα (Cat.No. ab24509, dilution 1:2,000) and UCP1 (Cat.No. ab10983, dilution 1:10,000) were obtained from Abcam (Abcam plc., Cambridge, UK). PGC-1α was purchased from Santa Cruz (Santa Cruz Biotechnology, Cat.No. sc-13067, 1:1,000). Antibody against GAPDH (Cat. No. GTX100118, dilution 1:10,000) was obtained from GeneTex (GeneTex Inc., CA, USA). The primary antibodies were prepared in 3% bovine albumin in Tris-buffered saline containing 0.1% (v/v) Tween 20 (TBST) and 0.05% (m/v) sodium azide with dilution factors as indicated. The anti-mouse and anti-rabbit secondary antibodies conjugated with horseradish peroxidase (HRP) were purchased from Thermo Scientific (Thermo Fisher Scientific Inc., MA, USA) and were diluted 10,000 times in 3% non-fat dry milk dissolved in TBST. Carvedilol active pharmaceutical ingredient (HPLC assay 100%) (Batch No. 16CI000002) was obtained from CTX Lifescience (CTX Lifescience Pvt Ltd., Surat, India). Protease inhibitor tablets were purchased from Thermo Fisher Scientific (Thermo Fisher Scientific Inc., MA, USA) and phosphatase inhibitor cocktail tablets were purchased from Roche (F. Hoffmann-La Roche Ltd., Basel, Switzerland). D-glucose was purchased from Intron Biotechnology (Intron Biotechnology Co., Ltd., Gyeonggi-do, Korea) and sodium pyruvate (Product. No. S024) was purchased from Toku-E (WA, USA). Bovine serum albumin (Product No. MB083) was purchased from Himedia (Mumbai, India). All other reagents were purchased from Sigma-Aldrich unless otherwise stated.

### Animals

All animal experiments in this study were approved by the Institutional Animal Care and Use Committee (IACUC) of the Tan Tao University, School of Medicine (Certificate No. 02/ 2018-HDKH.TTU). Swiss mice were purchased from Pasteur Institute (Ho Chi Minh city, Viet Nam). The mice were kept in controlled room temperature (22 ± 1˚C) with a 12-hour-light/dark cycle (light on/off at 06:00 a.m./p.m.). Mice were fed a normal chow diet (AniFood, Pasteur Institute-VN, 3.84 kcal/kg with 6–8% kcal from fat, NC) or a high fat diet (Research Diets D12492, USA, 5.24 kcal/kg with 60% kcal from fat, HFD) with filtered water provided *ad libitum*. After 2-week period of HFD feeding, 6-week-old male mice were administered 30 mg/ kg body weight of carvedilol dissolved in 0.001M acetic acid, pH 3.8 (carvedilol treatment group, body weight 24.14 ± 0.70 grams, N = 6) or 0.001M acetic acid (vehicle controlled group, body weight 24.96 ± 0.49 grams, N = 6) via oral gavage for 4-week treatment period. Body weight and food intake were monitored daily before treatment. Blood glucose levels (fed and fasted state) and tolerance tests were measured as indicated. After experimental period, mice were sacrificed by decapitation following ketamine anesthesia (100 mg/kg body weight) and the organs and plasma samples were collected and stored at -80 ᵒC for further analyses.

For studying the relationship between plasma leptin, insulin and norepinephrine levels, an independent cohort of 4-week-old male mice was divided into two groups and fed either NC or HFD for 8 weeks (6 mice per group, body weight 13.18 ±0.49 grams for NC-fed group and 12.86 ±0.45 grams for HFD-fed group). Blood samples were collected at the time points of 0, 4

and 8 weeks during the HFD challenging period in basal resting condition (resting state at day-time with food and water *ad libitum*).

## Measurement of blood glucose levels

Blood samples were taken from a tail nick and glucose levels were determined by the glucose oxidase method using a commercial blood glucometer (SAFE-ACCU, Shanghai International Holding Corp., Germany). For fed state blood glucose, mice were removed from food 2 hours before measurement. To determine fasted state blood glucose, mice were fasted overnight (16 hours) with water *ad libitum* provided.

## Glucose, insulin and pyruvate tolerance tests

Glucose (GTT), insulin (ITT) and pyruvate (PTT) tolerance tests were performed as described previously [37] after the mice were challenged with HFD and treated with carvedilol for 4 weeks. Body weight were 48.98 ± 1.22 g for HFD-fed + vehicle treatment cohort (n = 6) and 47.62 ± 0.91 g for HFD-fed + carvedilol treatment cohort (n = 6), P = 0.392. For GTT and PTT, mice were fasted overnight for 16 hours with water *ad libitum* provided. After measurement of fasted glucose levels, mice were intraperitoneally injected with a glucose solution (1.5 g/kg body weight) or sodium pyruvate solution (2 g/kg body weight) in normal saline. For ITT, mice were fasted for 2 hours and provided with water *ad libitum*. After measurement of basal glucose levels, 0.75 U/kg body weight of regular insulin (Humulin® U-100, Eli Lilly and Co., IN, USA) in normal saline was administered intraperitoneally. Blood samples were taken from a tail nick at 0, 15, 30, 60, 90, 120, and 150 minutes after injection and glucose levels were measured. The area under the curve (AUC) was determined to quantify the glucose, pyruvate and insulin tolerance.

## Blood collection and plasma preparation for hormones measurement

Blood (approx. 50 μl) samples were gently collected via a tail nick using commercial EDTA-coated tubes (Microvette® CB 300, Sarstedt, Germany) and then immediately centrifuged at 10,000 rpm, 4 °C for 10 minutes. The plasma supernatants were collected and stored at -80 °C before analyzed [38, 39].

## Leptin and insulin measurement

We measured plasma leptin and insulin by using ELISA kits (Cat. No. 90030 for leptin and Cat. No. 90080 for insulin, respectively, Crystal Chem, USA) in accordance with manufacturer's instructions.

## Norepinephrine measurement

Plasma norepinephrine was extracted, acylated, and then enzymatically converted before being quantitatively determined by a competitive enzyme immunoassay method using commercial ELISA kits (Cat. No. BA E-5200, Labor Diagnostika Nord GmbH & Co., Germany) followed the manufacturer's instructions as described previously [40].

## SDS-PAGE and Western blotting

Tissues from organs were homogenized using a glass Dounce homogenizer and lysed in RIPA buffer containing protease and phosphatase inhibitors. Total protein concentrations of lysed samples were determined by using Pierce™ Coomassive (Bradford) Protein Assay Kit (Thermo Fisher Scientific Inc., MA, USA). 20 μg of the protein lysates was electrophoresed on

SDS-PAGE gel and transferred onto nitrocellulose membranes. After blocking with 5% non-fat dry milk dissolved in TBST for 1 hour, the membranes were probed with a given primary antibody overnight at 4°C, reacted with HRP-conjugated goat anti-rabbit or anti-mouse IgG secondary antibody and were detected by using the Miracle Star[TM] Femto Western Blot Detection System (Intron Biotechnology Co., Ltd., Gyeonggi-do, Korea) and X-ray film (Ultra-Cruz® Autoradiography Film, Santa Cruz Biotechnology Inc., TX, USA) or iBright CL1000 Imaging System (Thermo Fisher Scientific Inc., MA, USA). Quantification of blots (densitometry) was performed using NIH ImageJ software.

## Histological analysis

White adipose tissue (WAT) and brown adipose tissue (BAT) samples were fixed in 4% formaldehyde diluted in phosphate-buffered saline (PBS) and sent to a medical center for histological analysis. Briefly, formalin-fixed tissues were dehydrated and embedded in paraffin. The paraffin-embedded sections were then cut into 4-μm slices and stained with hematoxylin and eosin (H&E). Stained slices were imaged by Dewinter Premium Digital Camera Microscope system (Dewinter Optical Inc., India). Quantification of adipocyte size from histological images was performed by an independent researcher using the NIH ImageJ Adipocytes Tool (https://github.com/MontpellierRessourcesImagerie/imagej_macros_and_scripts/wiki/Adipocytes-Tools).

## Statistical analysis

The GraphPad Prism 5.0 software was used for all statistical analyses. Two-way ANOVA with Bonferroni's post hoc tests or one-way ANOVA with Turkey's post hoc tests or Student's t tests were used to assess the statistical difference between groups as indicated. P < 0.05 was regarded as a statistically significant difference.

# Results

## Basal plasma norepinephrine was elevated and highly correlated with plasma leptin in HFD-induced obesity

Leptin has been shown to promote the synthesis and release of catecholamine [15, 17–20], possibly via modulating the expression/activity of tyrosine hydroxylase, the first and rate-limiting enzyme of the catecholamine synthesis [17, 21]. Plasma leptin concentration increases in proportion to the fat mass expansion in obesity which is frequently associated with sympathetic overactivation [2]. This led us to ask whether basal plasma norepinephrine, the principal catecholamine as an index of sympathetic activity [34], elevates and correlates with plasma leptin in obesity. To examine this possibility, we employed the HFD-induced obese mouse model to assess the relationship between plasma leptin and basal norepinephrine concentrations. As shown in Fig 1A and S1 Fig, 4-weeks of the HFD challenge was sufficient to induce a typical obese condition characterized by increased body weight (Fig 1A), increased blood glucose levels, impaired glucose and insulin tolerance and adipocyte enlargement (S1A–S1D Fig). Plasma leptin was markedly increased after 4 weeks of HFD feeding and further elevated thereafter (Fig 1B). Parallel with the increase in plasma leptin, plasma norepinephrine measured in the basal resting condition was significantly increased in the HFD-fed mice and remained at high levels during HFD challenge period (Fig 1C). Moreover, linear regression analysis revealed a high positive correlation [41] between basal norepinephrine and plasma leptin (r = 0.725, P < 0.0001, Fig 1D). Although plasma insulin was also gradually increased during the development of HFD-induced obesity, there was a low correlation [41] between basal norepinephrine

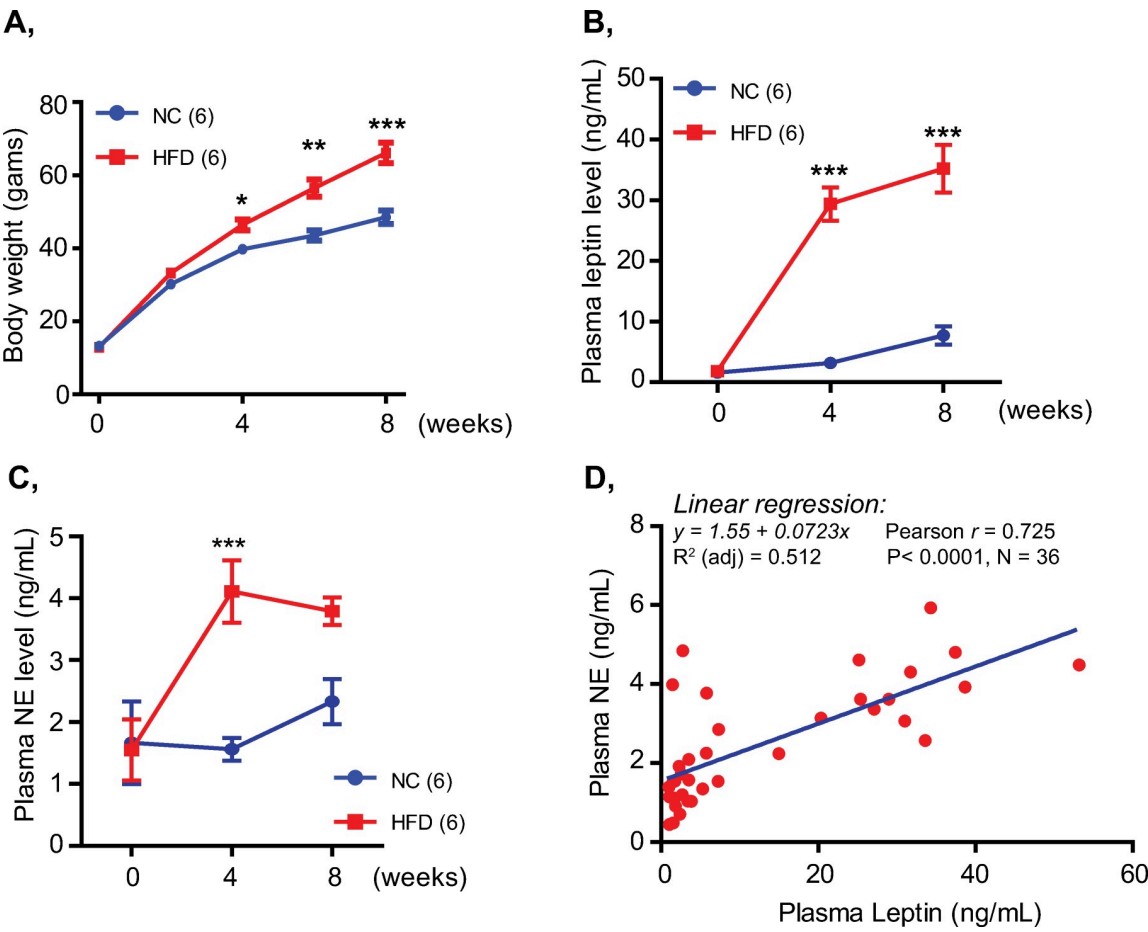

**Fig 1. Basal plasma norepinephrine was persistently elevated and highly correlated with plasma leptin in HFD-induced obese mice.** A, Body weight of male mice fed NC or HFD for 8 weeks. B, C, Plasma leptin (B) and norepinephrine (NE) (C) levels of mice measured in basal resting condition during 8-week period of HFD feeding. D, Linear correlation analysis of basal norepinephrine and plasma leptin concentrations during HFD-induced obesity. Data are presented as mean ± SEM. Two-way ANOVA with Bonferroni's post-tests. *$P< 0.05$, **$P<0.01$ and ***$P<0.001$. Hormone measurement was performed in triplicate.

and plasma insulin ($r = 0.428$, $P = 0.009$, S1E and S1F Fig). These results suggested that basal norepinephrine was persistently elevated during HFD-induced obesity development and highly correlated with plasma leptin, but not plasma insulin.

### Hepatic glucose overproduction and muscular insulin insensitivity associated with the adrenergic overdrive in HFD-induced obesity were attenuated by carvedilol treatment

Since catecholamine has broad and complex interactions on the glucose metabolism by exerting differing effects on metabolic organs including liver, muscle and adipose tissue [33], we next investigated whether chronic elevation of basal plasma norepinephrine led to an adrenergic signaling overactivation in these organs and altered metabolic functions of HFD-fed mice. As shown in Fig 2A, 2B, 2F and 2G and S2A and S2B Fig, significant increase in levels of p-Creb, a downstream effector of the β-adrenergic receptor/cAMP signaling pathway [42], in the livers, muscles, and adipose tissues of HFD-fed mice indicated an activation of adrenergic signaling pathway in these metabolic organs following the chronic elevation of basal

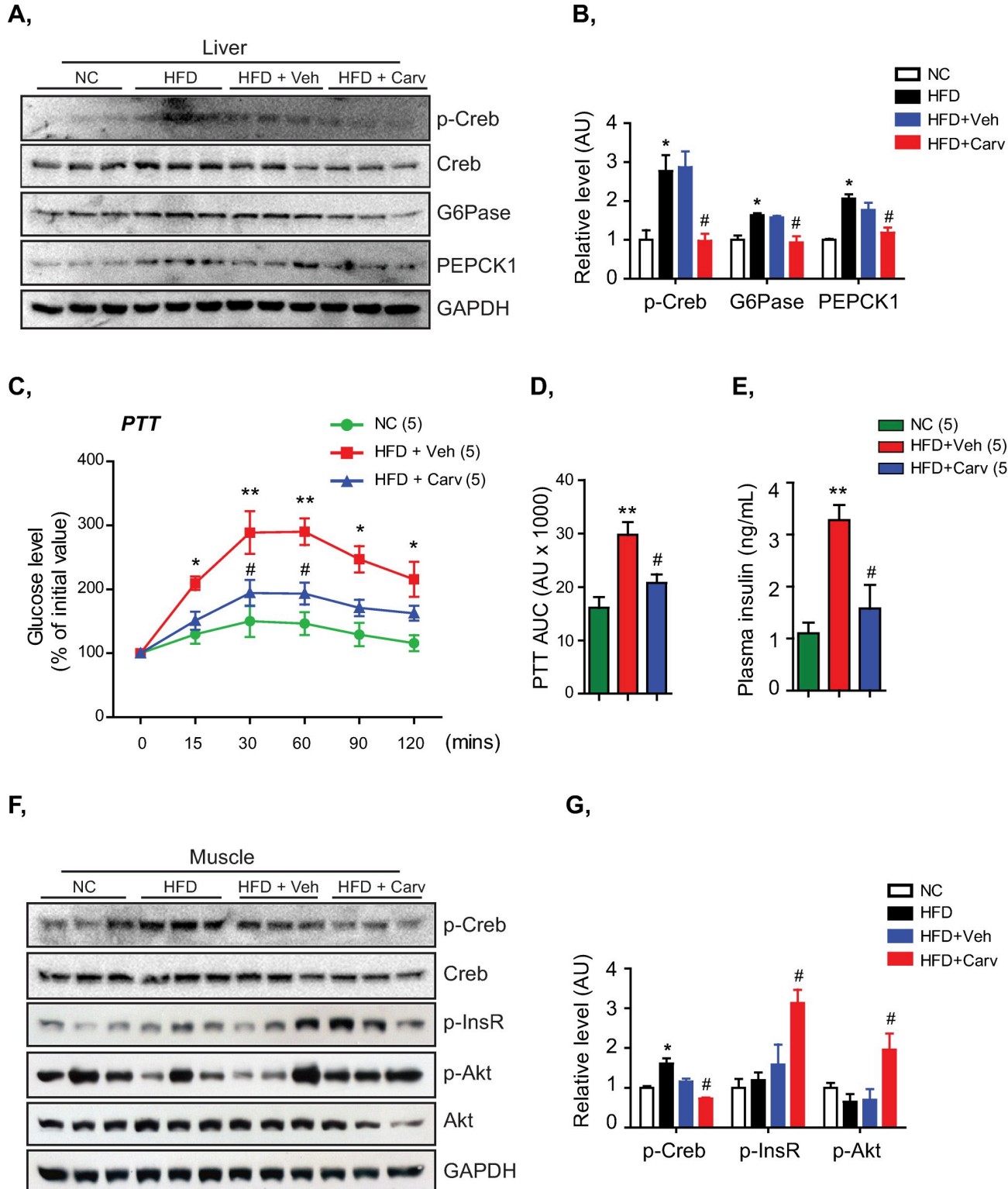

**Fig 2. Carvedilol treatment attenuated the hepatic glucose overproduction and muscular insulin signaling impairment associated with the HFD-induced adrenergic overdrive.** A, B, Immunoblots (A) and quantitative densitometry (B) showing the levels of p-Creb, G6Pase and PEPCK1 in the livers of NC-fed mice and HFD-fed mice treated with vehicle (Veh) or carvedilol (Carv). Normalized to total Creb or GAPDH. C, D, PTT (C) and PTT AUC (D) of NC-fed mice and HFD-fed mice treated with vehicle or carvedilol. E, Plasma insulin of NC-fed mice and HFD-fed mice treated with carvedilol or vehicle. F, G, Immunoblots (F) and quantitative densitometry (G) showing the levels of p-Creb, p-InsR and p-Akt in the muscles of NC-fed mice and

HFD-fed mice treated with vehicle (Veh) or carvedilol (Carv). Normalized to total Creb, Akt or GAPDH. Data are presented as mean ± S.E.M. One-way ANOVA with Turkey's post-tests in bar graphs and two-way ANOVA with Bonferroni's post-tests in line graphs. *$P < 0.05$, **$P < 0.01$ compared to NC group, #$P < 0.05$ compared to HFD+Veh group. Hormone measurement and Western blot analysis were performed in triplicate.

norepinephrine. Increased p-Creb levels were accompanied by a significant induction of the gluconeogenic enzymes, G6Pase and PEPCK1, in the livers of HFD-fed mice (Fig 2A and 2B). Consistent with increased G6Pase and PEPCK1 levels in the livers, these mice showed a hepatic glucose overproduction under pyruvate tolerance tests (PTT) compared to NC-fed mice (Fig 2C and 2D). Moreover, HFD-fed mice displayed higher plasma insulin levels without the corresponding increase of p-InsR and p-Akt levels in the muscles demonstrating a blunted muscular insulin signaling pathway (Fig 2E–2G).

We therefore designed an experiment to completely block the adrenergic overactivation in HFD-fed mice with carvedilol, a third-generation beta-blocker with additive $\alpha_1$-adrenoceptor antagonizing action [27, 28, 35, 36], to examine whether carvedilol treatment in HFD-induced obesity could target the adrenergic overdrive and provide metabolic benefits. As shown in Fig 2A, 2B, 2F and 2G, 30 mg/kg of body weight of carvedilol treatment completely blocked the increase of p-Creb levels in livers and muscles of the HFD-fed mice confirming the adrenergic signaling inhibition of carvedilol treatment [43]. Interestingly, carvedilol treatment was effective in suppressing the HFD-induced induction of gluconeogenic enzymes in the livers and partly corrected the hepatic glucose overproduction observed in HFD-fed mice (Fig 2A–2D). Moreover, carvedilol treatment lowered plasma insulin levels and significantly enhanced the muscular insulin signaling pathway of HFD-fed mice (Fig 2E–2G). These results indicated that carvedilol treatment targeting the adrenergic overdrive in HFD-induced obesity effectively blocked the induction of gluconeogenic enzymes to suppress a hepatic glucose overproduction and enhanced the muscular insulin signaling pathway.

Unlike the results observed in livers and muscles of the carvedilol-treated mice, we found that carvedilol failed to suppress the HFD-induced increase of p-Creb levels in the brown adipose tissues (BAT) of HFD-fed mice (S2A and S2B Fig). In addition, levels of PGC1α, PPARα, and UCP1 which are molecular clues of lipid catabolism in the BAT samples [44] were not different between HFD-fed mice treated with carvedilol and vehicle control (S2A and S2B Fig). Moreover, we did not observe a significant change in the adipocyte size of BAT samples between these mice (S2C Fig). These results suggested that carvedilol treatment might not affect the brown fat metabolism in HFD-fed mice.

## Carvedilol treatment during HFD-induced obesity reduced white adipose tissue enlargement and improved glucose tolerance and insulin sensitivity without affecting body weight and blood glucose levels

We next investigated whether the beneficial effects of carvedilol on gluconeogenic enzymes, hepatic glucose production and muscular insulin signaling pathway were translated to the metabolic benefits of carvedilol treatment in HFD-induced obesity. Indeed, carvedilol treatment did not affect the body weight and total daily food intake of HFD-fed mice (Fig 3A and 3B), however, the HFD-fed mice treated with carvedilol for 4 weeks showed reduced white adipose tissue (WAT) enlargement compared to the vehicle-treated mice (Fig 3C). Consistent with reduced WAT enlargement, plasma leptin levels of carvedilol-treated mice were significantly lower than those of controlled mice (Fig 3D).

Interestingly, we did not observe any differences in the blood glucose levels in both fed and fasted state between these mice (S3 Fig). However, HFD-fed mice treated with carvedilol showed improved glucose tolerance (Fig 4A and 4B) and insulin sensitivity (Fig 4C and 4D)

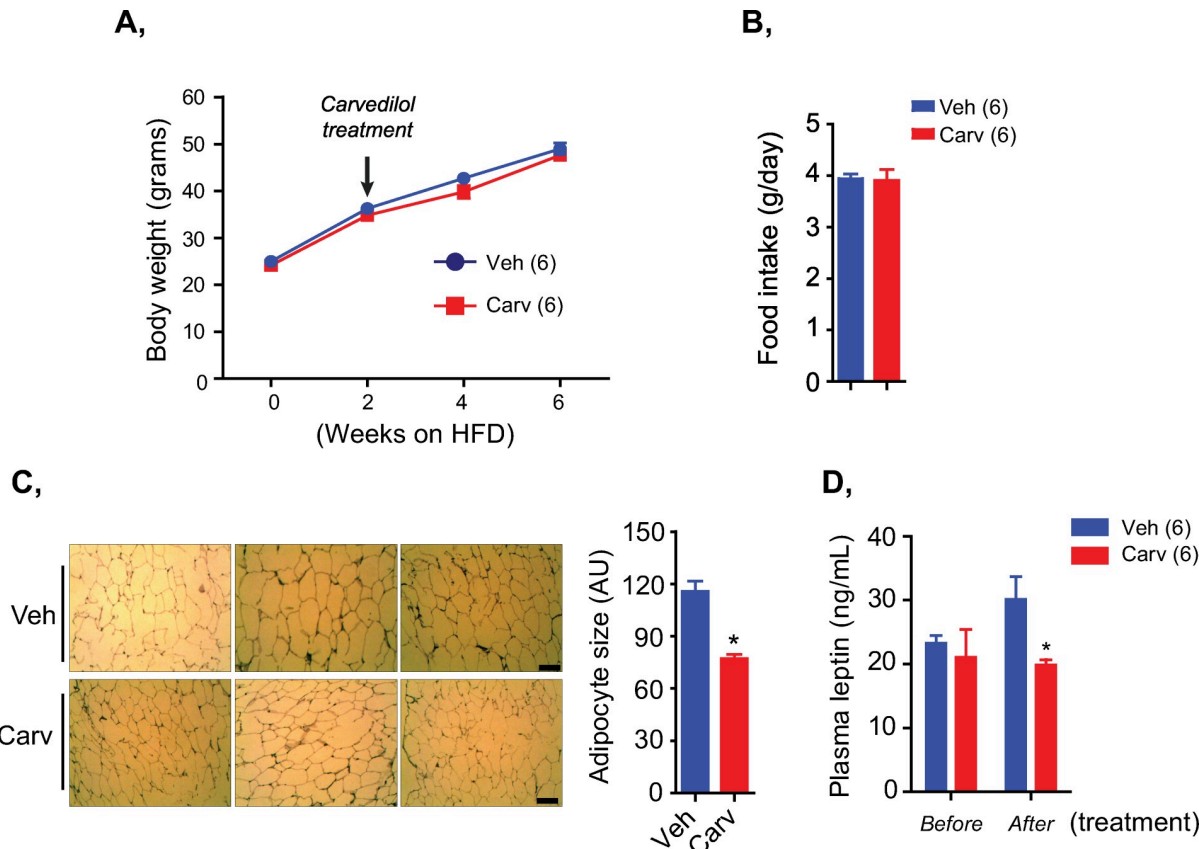

**Fig 3. Carvedilol treatment in HFD-fed mice reduced white adipose tissue enlargement and plasma leptin without affecting food consumption and body weight.** A, Body weight of HFD-fed mice treated with vehicle (veh) or carvedilol (carv). B, Total daily food intake of HFD-fed mice treated with vehicle (veh) or carvedilol (carv) measured in 3 consecutive days. C, Representative figures of H&E staining (left) and quantification of white adipocyte size (right) of HFD-fed mice treated with vehicle or carvedilol. Scale bar, 100 μm. D, Plasma leptin levels of mice before and after carvedilol treatment. Data are presented as mean ± S.E.M. Student's t-tests in bar graphs and two-way ANOVA in line graphs. Hormone measurement and histological analysis were performed in triplicate.

during glucose and insulin tolerance tests (GTT and ITT). These metabolic effects of carvedilol treatment could be attributed to its beneficial effects on the gluconeogenesis (Fig 2A–2D) and insulin signaling pathway (Fig 2E–2G), but not due to a change in energy consumption (Fig 3B). Taken together, these results suggested that carvedilol treatment targeting the adrenergic overactivation in HFD-induced obese mice helped to improve glucose tolerance and insulin sensitivity without affecting body weight and blood glucose levels.

## Discussion

Using the HFD-induced obese mouse model, we observed a chronic excess of the basal plasma norepinephrine along with increasing plasma leptin levels during obesity development. This persisted elevation of basal plasma norepinephrine further highlighted an over-activation of the SNS which has been previously documented in obesity [2]. Interestingly, basal norepinephrine was well-correlated with the plasma leptin, but not plasma insulin in obesity. The results in our study were consistent with recent findings that leptin, but not insulin, mediated the SNS-driven increase in heart rate and blood pressure associated with obesity [10]. Moreover, since leptin has been shown to directly stimulate the catecholamine synthesis and secretion [15, 17–20], there might be a direct link between increased basal

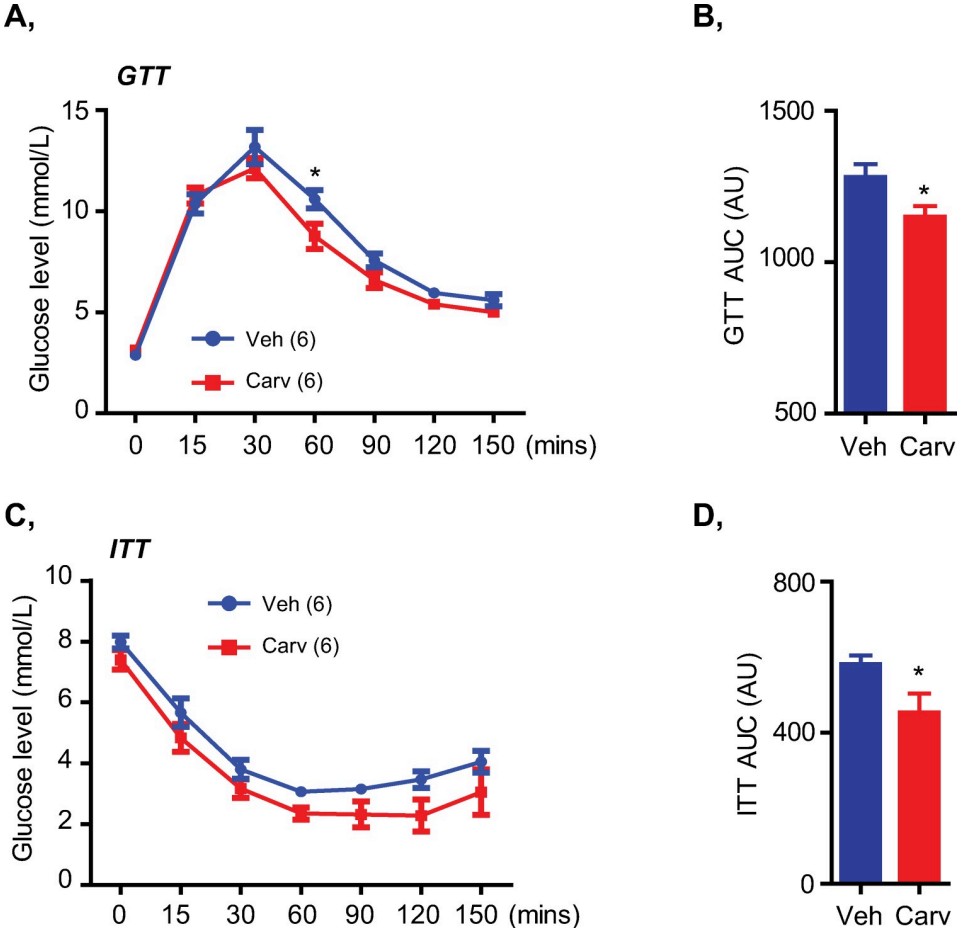

**Fig 4. Carvedilol treatment targeting the adrenergic overactivation in HFD-fed mice improved glucose tolerance and insulin sensitivity.** A, B, GTT (A) and GTT AUC (B) of HFD-fed mice treated with vehicle or carvedilol. C, D, ITT (C) and ITT AUC (D) of mice. Data are presented as mean ± S.E.M. Student's t-tests in bar graphs and two-way ANOVA with Bonferroni's post-tests in line graphs. *P< 0.05.

norepinephrine and the hyperleptinemic condition in obesity. However, further studies are required to confirm the causally direct relationship between basal norepinephrine and plasma leptin.

Excessive catecholamine representing an adrenergic hyperactivity is known to cause profoundly adverse effects on glucose metabolism [23, 25, 26, 45]. However, logically targeting the adrenergic overdrive in obesity by using adrenergic receptor antagonists in order to attenuate the metabolic disorders seems debatable since clinical evidence of the traditional beta-blockers showed negative effects on body metabolism possibly due to their incomplete adrenoceptor blockade properties [29, 30]. In the current study, we used carvedilol, a third-generation beta-blocker agent with vasodilating action [35, 36] which can completely block the effects of excessive catecholamine, to investigate whether therapeutically targeting the adrenergic overdrive from chronic excess of basal norepinephrine in obesity could lead to metabolic benefits on the glucose metabolism. In agreement with the results from recent clinical studies [31, 33, 46], we found that carvedilol treatment during HFD-induced obesity improved glucose tolerance and insulin sensitivity possibly by suppressing the hepatic glucose overproduction and enhancing the muscular insulin signaling pathway.

Unlike the results from a previous study showing that carvedilol had a hyperglycemic effect [47], our study showed that carvedilol treatment did not affect neither blood glucose levels nor body weight of the HFD-fed mice. Our findings of carvedilol's effects on blood glucose levels and glucose tolerance were consistent with the metabolic parameters of HFD-fed mice treated with carvedilol in a study of Wang *et al.* in which they used carvedilol to investigate diabetes-associated cardiac dysfunction [43]. The inconsistent findings on blood glucose of carvedilol treatment might be due to differences in the treatment time periods and/or the experimental rodent models since Wang *et al.* and our study used HFD-induced obese mice and treated carvedilol for 4 weeks [43] while Suresha *et al.* treated carvedilol in normal albino rats for only a 5-day period [47]. Nonetheless, our findings and others consistently indicated that the use of carvedilol in obesity might provide long-term benefits on glucose metabolism. Indirectly, these findings also further highlighted a previous notion that an obesity-associated sympathetic disorder as observed in our study by a chronic excess of basal norepinephrine in HFD-fed mice could, at least, partly contribute to the pathophysiology of metabolic disorders [6, 7].

Chronic adrenergic overdrive from excessive catecholamine might adversely affect the insulin sensitivity and glucose metabolism via complex interactions in several metabolic organs including increased lipolysis, impaired muscular glucose uptake, and increased hepatic glucose production [4–6]. By characterizing the primary targets of carvedilol treatment on suppression of hepatic glucose overproduction and enhancement of muscular insulin signaling, our study provided mechanistic insights into the metabolic benefits of carvedilol treatment in obese subjects. These metabolic benefits of carvedilol might be attributed to its ability to block the adrenergic overactivation in obesity since increased p-Creb levels in livers and muscles of the HFD-fed mice were totally blunted by carvedilol treatment. In contrast, the histological analysis and molecular markers of lipid metabolism in BAT samples were not changed by carvedilol treatment suggesting that effect of carvedilol on brown fat metabolism seemed minimal. Because carvedilol has been shown to have affinity with $\beta_3$-adrenoceptors in adipose tissues [48] and we found that the HFD-induced WAT enlargement was reduced by carvedilol treatment, our study therefore could not totally rule out carvedilol's actions in adipose tissues. In this regard, however, it would be interesting to examine whether carvedilol (and other third-generation beta-blockers with metabolic benefits) can specifically block the sympathetic hyperactivity to metabolic organs in different manners.

## Conclusions

The adrenergic overdrive which has been supposed to contribute to the pathogenesis of cardiovascular complications and metabolic disorders in obesity might directly connect to the increasing plasma leptin concentration [6–8]. Beta-blockers which can completely block the adrenergic overactivation might overcome the negative effects of traditional beta-blockers on glucose metabolism and insulin sensitivity [31]. In this regard, our study further provided an *in vivo* evidence of the relationship between chronic catecholamine excess and the hyperleptinemic condition in obesity and underpinned the view that therapeutic target of the chronic adrenergic overdrive in obesity by third-generation beta-blockers such as carvedilol might be helpful to attenuate obesity-related metabolic disorders.

## Supporting information

**S1 Fig. Metabolic characterization of HFD-induced obese mice and correlation of basal norepinephrine and plasma insulin levels.** A, Blood glucose levels at fed (left) and fasted state (right) of mice fed NC or HFD.
B, C, GTT (B) and ITT (C) of mice performed after 4 weeks feeding NC or HFD.

D, Representative figures of H&E staining (left) and quantification of adipocyte size (right) of white and brown adipose tissues of mice fed NC or HFD. Scale bar, 100 μm in WAT and 50 μm in BAT.

E, Plasma insulin levels of mice during 8-week period feeding NC or HFD.

F, Linear correlation analysis of basal plasma norepinephrine and insulin concentrations during HFD-induced obesity.

Data are presented as mean ± S.E.M. Student's t-tests in bar graphs and two-way ANOVA with Bonferroni's post-tests in line graphs. $^*P< 0.05$, $^{**}P<0.01$, $^{***}P<0.001$. Hormone measurement and histological analysis were performed in triplicate.

(PDF)

**S2 Fig. Molecular markers of fat metabolism and histological analysis of BAT samples of HFD-fed mice treated with carvedilol.** A, B, Immunoblots and quantitative densitometry (B) showing the levels of p-Creb, PGC1α, PPARα and UCP1 in BAT samples of NC-fed mice and HFD-fed mice treated with vehicle (Veh) or carvedilol (Carv). Normalized to total Creb or GAPDH.

C, Representative figures of H&E staining (left) and quantification of brown adipocyte size (right) of HFD-fed mice treated with vehicle or carvedilol. Scale bar, 50 μm.

Data are presented as mean ± S.E.M. Student's t-tests or one-way ANOVA with Turkey's post-tests in bar graphs. $^*P<0.05$. Western blot and histological analyses were performed in triplicate.

(PDF)

**S3 Fig. Carvedilol treatment did not affect blood glucose levels of HFD-fed mice.** Blood glucose levels at fed state (left) and fasted state (right) of HFD-fed mice treated with vehicle or carvedilol.

Data are presented as mean ± S.E.M. Student's t-tests.

(PDF)

**S4 Fig. Original uncropped and unadjusted images of blots.**

(PDF)

## Acknowledgments

We thank Mr. Nguyen Huu Dung (Pharmacist, R&D Boston Pharmaceutical Co., Viet Nam) for his generous providing of carvedilol active pharmaceutical ingredient. We are grateful to Mr. Isaac Smith (M.A, School of Humanities and Languages, Tan Tao University, Viet Nam) for his language editing of the manuscript.

## Author Contributions

**Conceptualization:** Ki Woo Kim, Khanh V. Doan.

**Formal analysis:** Linh V. Nguyen, Quang V. Ta, Ki Woo Kim, Khanh V. Doan.

**Funding acquisition:** Khanh V. Doan.

**Investigation:** Linh V. Nguyen, Quang V. Ta, Thao B. Dang, Phu H. Nguyen, Trang HT. Nguyen, Stephen Baker, Trung Le Tran, Dong Joo Yang, Ki Woo Kim.

**Methodology:** Linh V. Nguyen, Quang V. Ta, Thach Nguyen, Thi Van Huyen Pham, Stephen Baker, Dong Joo Yang, Ki Woo Kim, Khanh V. Doan.

**Project administration:** Khanh V. Doan.

**Supervision:** Stephen Baker, Ki Woo Kim, Khanh V. Doan.

**Validation:** Linh V. Nguyen, Quang V. Ta, Thach Nguyen, Ki Woo Kim, Khanh V. Doan.

**Writing – original draft:** Linh V. Nguyen, Quang V. Ta, Khanh V. Doan.

**Writing – review & editing:** Thao B. Dang, Thach Nguyen, Thi Van Huyen Pham, Trung Le Tran, Ki Woo Kim, Khanh V. Doan.

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
