## [Decision Letter · Decision Letter 0]

5 Sep 2019

PONE-D-19-21383

Carvedilol Improves Glucose Tolerance and Insulin Sensitivity in Treatment of Adrenergic Overdrive in High Fat Diet-induced Obesity

PLOS ONE

Dear Dr. Doan,

Thank you for submitting your manuscript to PLOS ONE. After careful consideration, we feel that it has merit but does not fully meet PLOS ONE’s publication criteria as it currently stands. Therefore, we invite you to submit a revised version of the manuscript that addresses the points raised during the review process.

We would appreciate receiving your revised manuscript by Oct 20 2019 11:59PM. To enhance the reproducibility of your results, we recommend that if applicable you deposit your laboratory protocols in protocols.io, where a protocol can be assigned its own identifier (DOI) such that it can be cited independently in the future. For instructions see: http://journals.plos.org/plosone/s/submission-guidelines#loc-laboratory-protocols

We look forward to receiving your revised manuscript.

Kind regards,

Dong-Gyu Jo, Ph.D

Academic Editor

PLOS ONE

Journal Requirements:

2. During your revisions, please note that a simple title correction is required: "Carvedilol Improves Glucose Tolerance and Insulin Sensitivity in Treatment of Adrenergic Overdrive in High Fat Diet-induced Obesity *in mice*". Please ensure this is updated both in the manuscript file and the online submission information.

3. To comply with PLOS ONE submissions requirements, in your Methods section, please provide additional information on the animal research and ensure you have included details on (i) methods of sacrifice, (ii) methods of anesthesia and/or analgesia, and (iii) efforts to alleviate suffering.

Reviewers' comments:

Reviewer's Responses to Questions

**Comments to the Author**

1. Is the manuscript technically sound, and do the data support the conclusions?

Reviewer #1: Yes

Reviewer #2: Yes

2. Has the statistical analysis been performed appropriately and rigorously? 

Reviewer #1: Yes

Reviewer #2: Yes

3. Have the authors made all data underlying the findings in their manuscript fully available?

Reviewer #1: Yes

Reviewer #2: Yes

4. Is the manuscript presented in an intelligible fashion and written in standard English?

Reviewer #1: Yes

Reviewer #2: Yes

5. Review Comments to the Author

Reviewer #1: In the manuscript entitled “Carvedilol Improves Glucose Tolerance and Insulin Sensitivity in Treatment of Adrenergic Overdrive in High Fat Diet-induced Obesity”, Nguyen et al. have investigated the metabolic effects of carvedilol, a third-generation beta-blocker agent, in the HFD-induced obese mice in which they found that the basal plasma norepinephrine of those mice is chronically elevated and correlated to the plasma leptin concentrations. They showed that treatment of adrenergic overactivation from the excessive norepinephrine in HFD-fed mice with carvedilol enhances glucose tolerance and insulin sensitivity by blocking the glucose overproduction in the liver and increasing muscular insulin sensitivity. Overall, the manuscript appears very solid and well done. The manuscript would benefit from addressing following points:

(1) The p-Creb level should be assessed together with total Creb level in addition to the loading protein in the samples.

(2) In Figure 2B, G and Figure S2B, the unit of Y-axis should be provided.

(3) In Figure S1D, the scale bar does not correspond to the absolute readings in the bar graphs. The authors should revise it.

Reviewer #2: This study submitted by Nguyen et al. claims that adrenergic overdrive such as high level of catecholamine link to plasma leptin level, consequently it contributes to the genesis of metabolic disorder. Nguyen et al. found that basal norepinephrine level is increased with a strong correlation of plasma leptin concentration in high-fat diet (HFD)-induced obese mouse model. By treating carvedilol, a third-generation beta type blocker, in HFD-induced obese mouse, the glucose tolerance and insulin sensitivity were improved.

This study is well designed and formed. Particularly Nguyen et al. carefully propose important point regarding potential causation of metabolic disorder which is the correlation between adrenergic overdrive and plasma leptin level, and they verified the effect of carvedilol against adrenergic overdrive.

I have an only minor comment regarding this study.

Figure legend should describe how many (N=??) repeat each experiment regardless of its type including western blot and its analysis. Although figure itself has displayed a number which I assume is an experimental number.

6. PLOS authors have the option to publish the peer review history of their article (what does this mean?). If published, this will include your full peer review and any attached files.

Reviewer #1: No

Reviewer #2: No

---

## [Author Response · Author response to Decision Letter 0]

26 Sep 2019

Rebuttal Letter (PONE-D-19-21383)

We sincerely appreciate the Academic Editor and Reviewers for taking your valuable time to consider our work. Here, we would like to address the journal requirements from the academic editor and the reviewer’s comments as follows:

Journal Requirements:

Response: 

We have formatted the manuscript following PLOS ONE's style requirements. Additionally, for this reason, we have rearranged the Figure 3 and Supplemental Figure 3 to make 04 main figures in the revised manuscript. We have therefore updated the Figs 3 and 4 and S3 Fig and the text in the “Results” and “Figure legends” sections in the revised manuscript correspondingly (lines 273-291, 537-553 and 584-587).

2. During your revisions, please note that a simple title correction is required: "Carvedilol Improves Glucose Tolerance and Insulin Sensitivity in Treatment of Adrenergic Overdrive in High Fat Diet-induced Obesity *in mice*". Please ensure this is updated both in the manuscript file and the online submission information.

Response:

We have corrected and updated the title “Carvedilol improves glucose tolerance and insulin sensitivity in treatment of adrenergic overdrive in high fat diet-induced obesity in mice” both in the revised manuscript (line 2) and the online submission information as the Academic Editor suggested.

We sincerely appreciate the Academic Editor for this insightful correction to better clarify our findings.

3. To comply with PLOS ONE submissions requirements, in your Methods section, please provide additional information on the animal research and ensure you have included details on (i) methods of sacrifice, (ii) methods of anesthesia and/or analgesia, and (iii) efforts to alleviate suffering.

Response:

We apologize the Academic Editor for the missing information of animal research on (i) methods of sacrifice, (ii) methods of anesthesia and/or analgesia, and (iii) efforts to alleviate suffering in the “Methods” section. We sacrificed mice by decapitation after anesthetizing with ketamine. We have provided this detail information in the “Methods” section in the revised manuscript (lines 131-132).

Response:

We have provided the original uncropped and unadjusted images of all blots/gels in the S4 Fig of the Supporting Information file. 

Reviewers' comments:

Reviewer's Responses to Questions

Comments to the Author

1. Is the manuscript technically sound, and do the data support the conclusions?

Reviewer #1: Yes

Reviewer #2: Yes

Response:

We sincerely appreciate the reviewer’s comment and suggestion on our manuscript.

2. Has the statistical analysis been performed appropriately and rigorously?

Reviewer #1: Yes

Reviewer #2: Yes

Response:

We sincerely appreciate the reviewer’s comment and suggestion on our manuscript.

3. Have the authors made all data underlying the findings in their manuscript fully available?

Reviewer #1: Yes

Reviewer #2: Yes

Response:

We sincerely appreciate the reviewer’s comment and suggestion on our manuscript.

4. Is the manuscript presented in an intelligible fashion and written in standard English?

Reviewer #1: Yes

Reviewer #2: Yes

Response:

We sincerely appreciate the reviewer’s comment and suggestion on our manuscript.

5. Review Comments to the Author

Reviewer #1: 

In the manuscript entitled “Carvedilol Improves Glucose Tolerance and Insulin Sensitivity in Treatment of Adrenergic Overdrive in High Fat Diet-induced Obesity”, Nguyen et al. have investigated the metabolic effects of carvedilol, a third-generation beta-blocker agent, in the HFD-induced obese mice in which they found that the basal plasma norepinephrine of those mice is chronically elevated and correlated to the plasma leptin concentrations. They showed that treatment of adrenergic overactivation from the excessive norepinephrine in HFD-fed mice with carvedilol enhances glucose tolerance and insulin sensitivity by blocking the glucose overproduction in the liver and increasing muscular insulin sensitivity. Overall, the manuscript appears very solid and well done. The manuscript would benefit from addressing following points:

(1) The p-Creb level should be assessed together with total Creb level in addition to the loading protein in the samples.

Response:

We agree with the reviewer that the p-Creb level should be measured together with total Creb. We have therefore run the samples to check total Creb levels and provided the results in the Figs 2A, F and S2A Fig in the revised manuscript. Similar to the loading protein GAPDH in the samples, the levels of total Creb were comparable between control and carvedilol treatment groups. Further, we have updated the Figs 2A-B, F-G and S2A and B Fig using the ratios of p-Creb/total Creb correspondingly. We thank the reviewer for this insightful suggestion.

(2) In Figure 2B, G and Figure S2B, the unit of Y-axis should be provided.

Response:

We apologize the reviewer for the missing information on the unit of Y-axis in the Figure 2B, G and Figure S2B. We have updated the Fig 2B, G and S2B Fig providing the unit of Y-axis in the revised manuscript. We sincerely thank the reviewer for this comment.

(3) In Figure S1D, the scale bar does not correspond to the absolute readings in the bar graphs. The authors should revise it.

Response:

We apologize the reviewer for the confusing correspondence between scale bar and absolute readings in the bar graphs in Figure S1D. We have revised the quantification of histological images and updated the results in the S1D Fig in the revised manuscript correspondingly. We sincerely appreciate the reviewer for this insightful comment to improve our manuscript.

Reviewer #2: 

This study submitted by Nguyen et al. claims that adrenergic overdrive such as high level of catecholamine link to plasma leptin level, consequently it contributes to the genesis of metabolic disorder. Nguyen et al. found that basal norepinephrine level is increased with a strong correlation of plasma leptin concentration in high-fat diet (HFD)-induced obese mouse model. By treating carvedilol, a third-generation beta type blocker, in HFD-induced obese mouse, the glucose tolerance and insulin sensitivity were improved.

This study is well designed and formed. Particularly Nguyen et al. carefully propose important point regarding potential causation of metabolic disorder which is the correlation between adrenergic overdrive and plasma leptin level, and they verified the effect of carvedilol against adrenergic overdrive.

I have an only minor comment regarding this study.

Figure legend should describe how many (N=??) repeat each experiment regardless of its type including western blot and its analysis. Although figure itself has displayed a number which I assume is an experimental number.

Response:

We apologize the reviewer for missing information on the experimental replication in the figure legends. We performed the experiments of hormone measurement, histological and western blot analyses in triplicate. We have updated the figure legends to describe information on the replication of each experiment in the revised manuscript correspondingly (lines 518, 534-535, 546, 569-570 and 581-582). We appreciate the reviewer for this insightful and constructive suggestion.

6. PLOS authors have the option to publish the peer review history of their article (what does this mean?). If published, this will include your full peer review and any attached files.

Do you want your identity to be public for this peer review? For information about this choice, including consent withdrawal, please see our Privacy Policy.

Reviewer #1: No

Reviewer #2: No

Response:

We sincerely appreciate the reviewer’s comment and suggestion on our manuscript.

We sincerely thank the reviewers for the careful reading and insightful comments on our manuscript.

Sincerely,

Khanh V. Doan, Ph.D.,

Department of Pharmacology,

School of Medicine, Tan Tao University.

Address: Tan Tao University Ave., Tan Duc E.City, Duc Hoa, Long An 850000, Viet Nam. 

Tel: (+84-272)-376-9216

Fax: (+84-272)-376-9208

Cell: (+84-91)-185-1177

Email: doankhanh.pharm@gmail.com

---

## [Editor Report · Decision Letter 1]

21 Oct 2019

Carvedilol improves glucose tolerance and insulin sensitivity in treatment of adrenergic overdrive in high fat diet-induced obesity in mice

PONE-D-19-21383R1

Dear Dr. Doan,

We are pleased to inform you that your manuscript has been judged scientifically suitable for publication and will be formally accepted for publication once it complies with all outstanding technical requirements.

With kind regards,

Dong-Gyu Jo, Ph.D

Academic Editor

PLOS ONE
---

## [Editor Report · Acceptance letter]

24 Oct 2019

PONE-D-19-21383R1 

Carvedilol improves glucose tolerance and insulin sensitivity in treatment of adrenergic overdrive in high fat diet-induced obesity in mice 

Dear Dr. Doan:

I am pleased to inform you that your manuscript has been deemed suitable for publication in PLOS ONE. Congratulations! Your manuscript is now with our production department. 

With kind regards,

on behalf of

Dr. Dong-Gyu Jo 

Academic Editor

PLOS ONE